# The Collegiate Athlete Perspective on Return to Sport Amidst the COVID-19 Pandemic: A Qualitative Assessment of Confidence, Stress, and Coping Strategies

**DOI:** 10.3390/ijerph19116885

**Published:** 2022-06-04

**Authors:** Oscar Levine, Michael Terry, Vehniah Tjong

**Affiliations:** Department of Orthopedic Surgery, Feinberg School of Medicine, Northwestern University, 676 North St. Clair Street, Suite 1350, Chicago, IL 60611, USA; m-terry@northwestern.edu (M.T.); vehniah.tjong@nm.org (V.T.)

**Keywords:** COVID-19, student athletes, college, return to sport, confidence, mental health, stress, coping, qualitative, interview

## Abstract

(1) Background: The COVID-19 pandemic has created challenges for college athletes as they return to sport and campus life. Emerging literature highlights some of these challenges, but no studies have used a primarily qualitative approach to assess the impact of the pandemic on college athletes. The purpose of this study was to better understand factors affecting college athletes’ return to sport and campus life amidst the COVID-19 pandemic. (2) Methods: Semi-structured interviews were conducted with varsity athletes who participated in the 2020–2021 season at a single university. Qualitative analysis was performed using the Strauss and Corbin method to derive codes, categories, and themes. Additionally, Athletic Coping Skills Inventory-28 (ACSI-28) scores were recorded and analyzed using descriptive statistics. (3) Results: A total of 20 student athletes were interviewed, revealing that confidence and motivation, increased stress and anxiety, and adaptive coping strategies were common themes affecting their experiences returning to sport and campus life. Results from the ACSI-28 showed an average score of 49.5 and a range of 24–66. (4) Conclusions: Semi-structured interviews revealed factors impacting return to sport and student life. These findings can help inform how to better support college athletes throughout the remainder of the current pandemic and moving forward.

## 1. Introduction

The COVID-19 pandemic has presented numerous life stressors for collegiate student athletes as they resume their seasons and adapt to an altered life on campus. According to a series of surveys administered to college athletes by the NCAA, the pandemic has brought about increased rates of mental health concerns, barriers to athletic training, challenges associated with a move to online learning, and the cancellation of championships [1,2]. Additional studies examining the impact of COVID-19 on the lives of athletes in general have identified similar stressors, including isolation from teammates and coaches, fear of contracting COVID-19, and reduced financial security [3] as well as the closure of training centers and the cancellation of competitions [4]. While these studies begin to characterize the impact of COVID-19 on the lives of college athletes and athletes in general, obtaining a more detailed understanding of COVID-19′s impact on collegiate student athletes remains an important task.

One potential way to gain this understanding is a qualitative, interview-based approach. Previous qualitative studies have demonstrated how various patient-derived themes from semi-structured interviews are associated with a more or less successful return to sport following orthopedic surgery [5,6,7,8]. Such themes associated with difficulty in returning to sport included competing life interests and fear of reinjury [5,7], while factors associated with a more successful return to sport included self-efficacy, social support, and resetting expectations [6]. Two recent studies used qualitative methods to better understand the impact of COVID-19 on elite athletes [9,10]. However, there are no studies to date that use qualitative methods to examine the impact of COVID-19 on collegiate student athletes specifically.

Collegiate student athlete mental health is an area of particular interest in this study given the need for a greater recognition of mental health concerns among college athletes [11]. Prior studies have demonstrated a prevalence of depression among college athletes between 15.6% and 21% [12]. Certain populations appear to be more affected than others; female gender and freshman status are associated with increased odds of experiencing depressive symptoms [11], and depression rates are higher in current as compared to former college athletes [13]. Several factors may contribute to these rates, including injury, over-training, involuntary career termination, and performance expectations [12]. Moreover, many student athletes do not seek mental health support due to stigma or a lack of mental health resources [14]. In addition to these concerns, existing literature points to an increase in negative emotions during the pandemic among both college students [15] and athletes in general [16]. Given the association of mental health concerns with COVID-19 and dual student and athlete identities, one goal of the present study is to better understand factors affecting the mental health of collegiate student athletes.

Adaptive coping strategies have the potential to protect against the added stressors brought on by COVID-19. Two quantitative studies conducted among collegiate student-athletes during COVID-19 indicated that high levels of social support were linked to better maintenance of athlete identity and mental health [17,18]. Semi-structured interviews with college athletes would allow for a deeper understanding of the different coping strategies they have used throughout the pandemic. In addition to eliciting coping strategies through semi-structured interviews, quantitative, survey-based measures can aid in understanding student athlete coping. One study investigating the impact of coping on recovery from ACL reconstruction found that lower scores on the Athletic Coping Skills Inventory-28 (ACSI-28) were significantly associated with a slower recovery from ACL reconstruction [19]. Together, qualitative and quantitative measures of coping have the potential to yield insights into individual differences and trends among collegiate student-athletes.

The present study aims to derive themes surrounding collegiate student athlete return to sport and student life during the COVID-19 pandemic using primarily qualitative (semi-structured interviews) as well as quantitative (athlete-specific coping strategies survey) measures. Examining player-derived themes relating to COVID-19 will contribute to a better understanding of how to better support college athletes during the remainder of this pandemic and moving forward.

## 2. Materials and Methods

### 2.1. Participants

Varsity athletes aged 18 or older at a single NCAA Division I school during the 2020–2021 season, including those who were on injured reserve, were eligible for the study.

### 2.2. Recruitment and Data Collection

An email invitation to participate in the study was sent to all 502 student athletes at Northwestern University over a period spanning from January to August 2021. The recruitment email contained the details of the study and a link to a survey administered on REDCap, which contained consent and demographics forms as well as the ACSI-28 survey. ACSI-28 scores served as a quantitative measure of student-athlete coping. The ACSI-28 is a validated, 28-question survey spread evenly across 7 subscales: coping with adversity, peaking under pressure, goal setting/mental preparation, concentration, freedom from worry, confidence and achievement motivation, and coachability [20]. The questions are scored on a 4-point Likert scale and are summed to determine the overall score [19]. Upon consenting and completing these surveys, student athletes were contacted via email about scheduling an interview.

Twenty- to thirty-five-minute audio-recorded telephone interviews were conducted by the lead author (O.P.L.) using a study-specific interview guide. This interview guide contained open-ended questions along with more specific probe questions. It was developed using previous qualitative studies regarding return to sport following orthopedic surgery [5,6] along with considerations specific to collegiate student-athletes and the COVID-19 pandemic. The interviewer used the method of active passivity to actively follow-up with comments made by the subjects while enabling them to respond freely to questions without intrusion [5,6,8]. An example of a question asked is, “What factors make you more or less confident in your return to play?” A possible follow-up question to this one is, “How has your ability to train throughout the pandemic been affected?

Data saturation was reached when the addition of another student athlete interview did not provide any additional new content. Once saturation was reached, additional players were no longer sought for interview.

### 2.3. Data Analysis

Interviews were analyzed by two researchers (O.P.L. and V.K.T.) using the Strauss and Corbin method of open coding, axial coding, and selective coding [21]. Paraphrased responses were first coded. Codes were then grouped into categories based on their commonalities, and these categories were ultimately grouped into overarching themes. ACSI-28 scores were analyzed using descriptive statistics.

## 3. Results

### 3.1. Characteristics of the Study Population

Overall, 502 student-athletes were contacted for the study, and 32 (6.4%) agreed to participate in the interviews. In total, 20 student athletes (4.0%) were interviewed and completed the ACSI-28 survey (0 dropped out). The average age of athletes studied was 20 years (18–23 years), with 85% being female. The majority had multiple years of eligibility remaining, and only 20% were in their senior year of study. Field hockey, dive, and cross country/track and field were the most represented sports in this study. Eight student athletes (40%) reported having an injury at some point during the season, and four (20%) reported injured-reserve status (Table 1).

### 3.2. Themes

Three key themes were used to describe student athletes’ experience with returning to sport during the COVID-19 pandemic: confidence and motivation, increased stress and anxiety, and adaptive coping strategies. Table 2 provides a list of common responses grouped by theme along with their corresponding frequency.

#### 3.2.1. Confidence and Motivation

Student athletes commonly cited the ability or inability to train as a factor impacting their confidence and motivation in returning to sport. Student athletes often noted a lack of access to athletic facilities, particularly during the summer offseason, as a factor hurting their physical preparedness and competition readiness coming into the season: “Having to go home and not having the equipment that Northwestern has…definitely worsened my shape” (S1); “I don’t have a gym or anything in my house, so it was kind of difficult to lift weights” (S2); “We were all a little nervous coming back, because some of us hadn’t been able to train…so we were all at different levels coming back” (S14). However, several student athletes reported that they felt just as prepared or more by the time their seasons started due to the extended pre-season: “If anything, we were almost more prepared because we only focused on ourselves and practiced against ourselves” (S4); “It was almost nice to build strength early with the off-season training” (S8); “As a freshman who was expected to start, I felt like I benefitted greatly from that time in the fall to train and kind of get used to the gym and get used to my role” (S10). In some cases, the extra time off from competition benefitted student athletes by taking the pressure off (S14), providing them with a chance to recover from injury (S9), or giving them a new appreciation for the sport (S5, S9, S10, S11, S14, S15).

Another major factor impacting student athlete confidence and motivation in returning to sport was the uncertainty surrounding the status of their season: “The motivation was pretty low, especially since we weren’t sure when we would be in competition” (S7); “The days got pretty monotonous because there wasn’t an end goal or goals along the way” (S13). For members of one team, several mid-season suspensions of play hurt their competitive rhythm: “It was a lot more stopping and starting, which was difficult physically and emotionally” (S8); “It was really hard to stay motivated and convince myself that I was going to get another opportunity because I was afraid it would take me a few days to get back to where I left the gym” (S10). Changes in training due to season uncertainty were also perceived to have contributed to injury in one student athlete (S18).

COVID-19 precautions also played an important role in return-to-sport confidence and motivation. While some student athletes reported that the masks made training less enjoyable (S1, S3, S9, S11, S18, S19), hurt motivation (S7), or hurt communication (S10), others got used to them over time (S7, S14), were relatively unaffected by mask wearing (S4, S8, S11, S15), or found them reassuring (S9). Although the majority of subjects were not concerned about contracting COVID-19 from their teammates, many were concerned about contracting it during travel (S1, S2, S4, S6, S8, S12, S16) or competition (S2, S7, S9, S11, S14, S20). One subject explained, “I think I was more worried about COVID than I was about doing well in meets because there were a lot of other teams who weren’t as safe and whose programs had been shut down multiple times” (S11).

#### 3.2.2. Increased Stress and Anxiety

A major source of stress and anxiety for the student athletes was the limitation placed on social life. The majority of subjects reported that their teams had a set of rules or expectations limiting the number of people they could come into contact with. As a result, some subjects reported a sense of heightened pressure to uphold their teams’ protocols: “Feeling that if I made any small error there would be big consequences just added a baseline level of stress” (S17); “[I felt] guilty if I ever wanted to hang out with members of my team that I didn’t live with” (S18). The lack of opportunities to socialize also made it harder on student athletes to relieve stress. As one subject noted, “Athletes in general, and myself included, put a ton of effort into our sport and to school as well, so we need an outlet” (S4). Several student athletes reported that they felt more connected to their teammates due to the social bubble but desired more connection with people outside of the team: “Yes it did make our team closer, but I also think it’s healthy to hang out with people outside of the team” (S14); “I think we were very much more connected this year, which is a good and bad thing, because now I feel very close to them, but sometimes if you’re with the same group for a long time, there are some habits of theirs that you get a little fed up with” (S11).

The change to online classes due to the pandemic was also a source of stress for student athletes. Subjects reported it causing more difficulty in focusing (S1, S2, S6, S9, S11, S18, S19, S20) and making it harder to build relationships in class (S8, S18) and harder to ask questions or reach out for help (S16, S18). Additionally, several subjects mentioned that office hours were more awkward or less appealing to attend (S1, S3, S7, S9). As a result of these changes, student athletes felt more isolated in their learning experience: “It’s a lot more self-teaching” (S1, S14). On the other hand, subjects noted various benefits associated with online learning, including saving time (S8, S16), providing flexibility during travel (S12), and allowing for students to learn at their own pace (S7).

Daily COVID-19 testing was both a source of anxiety and reassurance for student athletes. There was a common fear associated with the thought of testing positive for COVID-19: “There was a lot of anxiety when we were waiting for results because I never wanted to be that person who shut down our team” (S18); “Just being that person to shut down practice, I would feel so guilty” (S20). A number of subjects indicated that taking extra time out of the day to get tested added stress: “The fact that it takes another 15 min out of every day definitely contributes to [stress]” (S1); “I literally plan my days around the COVID testing windows” (S13). Despite these concerns, student athletes generally indicated that the daily COVID-19 testing provided reassurance: “It was a little bit stressful, but it was reassuring” (S19); “It is annoying having to take time out of my day to do it, but I understand why it needs to be done…It is definitely nice to know that I don’t have it” (S7); “I really liked [the antigen testing] because it made me feel more comfortable…knowing that every single person is taking care of themselves to the best of their ability” (S14).

#### 3.2.3. Adaptive Coping Strategies

Social support was one of the main adaptive coping strategies that student athletes cited as helping them through the pandemic. Teammates were widely seen as a strong source of social support. Some excerpts point to the power of shared experience: “With the teammates, they’re going through the same thing…my roommate, for example, knows exactly what it feels like” (S8); “I think that during this time I’ve gotten a lot closer to my teammates and the people that are going through the exact same experience as I am” (S9). Coaches were also widely seen as a good source of support, and subjects particularly appreciated how they made various help resources available (S4), facilitated team meetings with a sports psychologist (S8), and promoted racial justice discussions (S10). However, some student athletes reported that coaches struggled to set realistic expectations for their players at times: “The coach kind of has his idea of the season, which is not aligned with everyone else’s goals during COVID” (S2); “I think they put a lot of pressure on us that way because we didn’t perform that well at Big Tens my freshman year, so they wanted to prove something to everybody this year, which definitely made it difficult because I was more nervous to tell them if I was feeling any aches and pains just because I didn’t want to disappoint them” (S18). Family was also widely cited as a strong source of support, and several student athletes brought up their disappointment over their family members not being able to show their support directly at competitions (S1, S4, S6, S18).

Student athletes commonly cited physical activity and their sport in particular as a helpful outlet. As one subject explained, “I can be having the worst day…Once you get on the court, it kind of all goes away. You just want to be a good teammate” (S8). Subjects also talked about how adopting new hobbies and routines helped them deal with stress: “I really found a love for painting…It reminded me of my own training” (S14); “I’ve started reading more since COVID…and that’s helped me take my mind off of things” (S1).

All subjects employed a variety of thoughts and attitudes that helped them cope with difficulties during the pandemic. Some adopted a mindset of gratitude: “I’m just grateful to have the opportunity to, you know, even be at college, or be able to play the sport” (S8); “There are a lot of people our age who have it worse than we do, so we are lucky that we get to compete” (S12); “Usually, I go through a list of things I’m thankful for” (S16). Several turned to an inner sense of resiliency: “I can always adapt and I can always work harder to eventually get what I want to get” (S14); “I don’t really give my excuses for anything” (S7); “If something didn’t go my way…I would think about all the ways I could respond and which would be the most effective” (S10). On the other hand, some responded to the challenges of the pandemic by managing their expectations and acknowledging factors outside of their control: “It’s a pandemic. There’s nothing I can do about it” (S1); “It’s ok to not feel ok right now” (S11); “If things can change the way they did, then they can change back” (S13).

### 3.3. Quantiative Measures of Student Athlete Coping

Scores from the ACSI-28 survey ranged from 24 to 66, with an average of 49.5. The averages of the subscale scores (minimum score of 0, maximum score of 12) were as follows: coping with adversity (6.6), peaking under pressure (6.8), goal setting/mental preparation (7.2), concentration (6.8), freedom from worry (4.9), confidence and achievement motivation (7.7), and coachability (9.5) (See Table 3). Subjects with higher overall ACSI-28 scores were more likely to demonstrate gratitude or inner resilience in their thought patterns, while subjects with lower ACSI-28 scores were less likely to demonstrate these thought patterns. For example, S10 (score: 58) said, “If something didn’t go my way…I would think about all the ways I could respond and which would be the most effective” (inner resilience), and S16 (score: 64) kept a daily journal of things she was grateful for. On the other hand, S13 (score: 24) experienced increased stress and demonstrated an external locus of control in her thought pattern (“If things can change the way they did, then they can change back”).

## 4. Discussion

The present study identified three key themes affecting collegiate student athletes in their return to sport and campus life: confidence and motivation, increased stress and anxiety, and adaptive coping strategies. These themes can provide insights on how to better support student athletes moving forward.

### 4.1. Confidence and Motivation

A key factor impacting student-athletes’ confidence in their return to play was their ability or inability to train. A narrative emerged wherein some student athletes felt less prepared when they came back to campus in the fall due to an inability to train yet felt more prepared by the time their seasons started due to the extended pre-season and associated ability to build both individual fitness and team chemistry. The relatively large impact of ability to train on confidence fits with the existing quantitative literature. According to a survey sent to Estonian elite athletes during the pandemic, the closure of training centers was cited as causing distress among 57% of the respondents [4]. This underscores the importance of finding creative ways to maintain student athlete engagement in training regimens during athletic facility closures [22] as well as an acknowledgment that student athletes will need time to reacclimate to team training before resuming competition. When considering the barriers to training among collegiate student athletes, it is also important to consider factors outside of the facility closures themselves; according to the NCAA Student-Athlete COVID-19 Well-being Survey in fall of 2020, fear of exposure to COVID-19, lack of motivation, feelings of stress or anxiety, and sadness or depression were identified as barriers among 38%, 24%, 17%, and 10% of respondents, respectively [2]. Finally, it is interesting to note that the extended time away from training and competition had some positive effects: increased recovery time from injury, reduced self-imposed pressure, and heightened appreciation for one’s sport. These benefits are echoed by another qualitative study, which found that the pandemic gave elite athletes time to reflect on the significance of their participation in competitive sport [10].

The uncertainty surrounding the status of student athletes’ seasons also had a major influence on their confidence and motivation in returning to play. In particular, student-athletes tended to feel less goal-directed in the absence of knowing what their competitive season would look like and whether conference or NCAA championships would occur. Although 55% of the participants in the NCAA Student-Athlete COVID-19 Well-being Survey of spring 2020 indicated that the NCAA communicated COVID-19 developments in a timely fashion, only 43% of winter-sport student athletes perceived the NCAA’s decision to cancel championships as fair [1]. This raises the importance of good communication among the NCAA, athletic conferences, athletic departments, coaches, and players moving forward.

Finally, concerns over contracting COVID-19 played a role in the confidence with which student athletes participated in competition. The most common complaint against mask-wearing was that it made participating in their sport less enjoyable. Indeed, mask wearing has been shown in randomized controlled trials to diminish exercise time [23], hurt cardiopulmonary measures of performance [23,24], and increase discomfort [24]. Yet, many of the student athletes interviewed were concerned about contracting COVID-19 during travel or competition against teams who were less diligent about following COVID-19 guidelines. This is in line with data from the NCAA Student-Athlete COVID-19 Well-being Survey (fall 2020) showing that Division I student athletes were more likely to feel that their friends on campus were taking social distancing guidelines seriously than their friends in other towns [2]. When weighing the costs and benefits associated with mask wearing across various athletic settings, it is therefore important to take into account not just the physical discomfort of wearing masks during athletic activity but also the peace of mind they may provide student athletes.

### 4.2. Increased Stress and Anxiety

Restrictions on social life, often instituted on a team-by-team basis, were a major factor increasing student athletes’ stress and anxiety. These restrictions not only contributed to a heightened pressure to uphold expectations but also limited student athletes’ ability to de-stress. These findings build off recent quantitative work demonstrating significant correlations between student athlete social support, athlete identity, and mental health during the COVID-19 pandemic [17,18]. Together, these findings underscore the need to balance concerns over student athlete health and availability with their basic social needs.

Another source of stress and anxiety was the move to online classes, with student-athletes generally finding it harder to focus and feeling more isolated in their educational experience. The NCAA Student-Athlete COVID-19 Well-being Surveys found that the majority of student athletes surveyed felt positive about their ability to pass their classes [1,2]. Participants in the present study similarly did not generally present concerns about their ability to pass their classes. Nevertheless, the feelings of isolation associated with education during the pandemic highlight that there is room for improvement when it comes to fostering academic collegiality and engagement throughout the student athlete population.

Daily COVID-19 testing produced a mixed effect on student athlete anxiety and stress levels, with some indicating that daily testing raised these levels, while others found it reassuring. One study of COVID-19-testing programs among PAC-12 student athletes found that daily antigen testing was comparable to two to three times a week of reverse transcription-polymerase chain reaction (RT-PCR) testing at detecting COVID-19 cases [25]. Future quantitative research could illuminate the impact of different COVID-19-testing strategies on student athletes’ mental health.

### 4.3. Adaptive Coping Strategies

The importance of social support to athlete identity and mental health [17,18] as well as return to sport following orthopedic surgery [5,6] have been noted. In the present study, social support, particularly from teammates who were going through the same experiences, was a commonly cited adaptive coping strategy. While coaches were also widely seen as a source of social support, some student athletes reported a mismatch between their expectations and their coaches’. Positive experiences with coaches were sometimes accompanied with coaches providing extra support resources, while negative experiences were associated with coaches having performance expectations that exceeded the players’ expectations. This highlights two ideas: (1) that coaches can play an important role in improving the access that their student-athletes have to various support resources and (2) that communication and empathy between student athletes and coaches in setting expectations is important.

Student athletes commonly turned to their sport as an outlet for stress while also using other forms of physical activity and new hobbies, such as reading, as coping strategies. These findings build off another qualitative study of athletes during the COVID-19 pandemic, which found that new or adapted routines allowed athletes to regain a sense of goal directedness that was lost with the lack of organized training or competition [10].

Finally, student athlete thoughts and attitudes played a role in helping them cope with the stresses of the pandemic. Specifically, three major thought patterns were elicited: gratitude (“I’m just grateful to have the opportunity to…even be at college, or be able to play the sport” (S8)); inner resiliency (“I don’t really give my excuses for anything” (S7)); and acceptance of factors outside of one’s control (“It’s a pandemic. There’s nothing I can do about it” (S1)). The beneficial impact of resilient thinking is supported by previous qualitative studies that identified resiliency as a factor associated with successful return to sport following orthopedic procedures [5,6]. Furthermore, the positive impact of these thoughts can potentially be leveraged through journaling or more formal interventions, such as written emotional disclosure (WED) [26].

### 4.4. Secondary Outcomes

The ACSI-28 provided additional insight into the coping skills of student athletes during the pandemic. Of note, coachability had the highest mean subscale score (9.5), while freedom from worry had the lowest (4.9), reflecting common strengths and weaknesses, respectively, among the student athletes. Additionally, while there is not an established ACSI-28 cutoff score delineating adaptive versus maladaptive coping, a previous study looking at recovery from ACL surgery found that individuals who scored lower than 58 took two months longer on average to recover from surgery than individuals who scored 58 or higher [19]. The seven subjects with scores greater than or equal to 58 (S4, S5, S7, S8, S10, S14, and S16) all demonstrated either gratitude or inner resilience in their thought patterns, while subjects who scored below 58 were less likely to employ such thoughts. Future studies could further elucidate correlations between ACSI-28 scores and qualitative reports of coping strategies.

### 4.5. Limitations

Although qualitative methods have the advantage of being able to generate subject-derived themes, they also have several limitations. The fact that data saturation was defined by the research team introduced interviewer and responder bias. Additionally, there was volunteer bias, whereby the subjects who agreed to participate in the study likely shared certain characteristics. In our particular sample, 85% of the participants were female. Moreover, the small size limited the representativeness of the sample. For example, there was a lack of representation from a number of sports teams, such as football. The study population was limited to a single Division I NCAA institution, which limits the generalizability of the study. Given these limitations, future studies examining the impact of COVID-19 on college athletes could aim for a larger, more representative sample that includes multiple collegiate athletic programs.

## 5. Conclusions

COVID-19 has created a challenging environment for collegiate student athletes, adding stressors to the multiple responsibilities they already face. Through semi-structured interviews, themes relating to confidence and motivation, increased stress and anxiety, and adaptive coping strategies were identified as factors affecting the lives of student athletes both on and off the field. Along with prior studies, the insights gained from this study can inform how to better support collegiate student athletes throughout the remainder of the current pandemic and moving forward.

## Figures and Tables

**Table 1 ijerph-19-06885-t001:** Student Athlete Demographics.

Variable		Total (*n* = 20)
Age, y (mean, (range))		20 (18–23)
Year (*n*, (%))		
	1st	5 (25)
	2nd	6 (30)
	3rd	5 (25)
	4th	2 (10)
	5th plus	2 (10)
Sex (*n*, (%))		
	Female	17 (85)
	Male	3 (15)
Sport (*n*, (%))		
	Field Hockey	4 (20)
	Fencing	1 (5)
	Cross Country/Track and Field	4 (20)
	Lacrosse	1 (5)
	Baseball	1 (5)
	Volleyball	2 (10)
	Dive	4 (20)
	Golf	1 (5)
	Softball	1 (5)
	Soccer	1 (5)
Injured (*n*, (%))		8 (40)
Status		
	Active	16 (80)
	Injured Reserve	4 (20)

**Table 2 ijerph-19-06885-t002:** Common Responses Listed by Theme.

Response by Theme	Number of Responses (%)
**Confidence and Motivation**	
Inability to train led to feeling less ready	10 (50)
Extended pre-season led to more readiness	8 (40)
Uncertainty hurt motivation	9 (45)
Masks hurt training	8 (40)
Masks did not impact training/were reassuring	5 (25)
Concerned about contracting COVID during travel/competition	12 (60)
**Increased Stress and Anxiety**	
Social restrictions contributed to stress	15 (75)
Change to online classes contributed to stress	12 (60)
Change to online classes provided flexibility	4 (20)
Daily COVID testing contributed to stress	9 (45)
Daily COVID testing provided reassurance	8 (40)
**Adaptive Coping Strategies**	
Social support from teammates	12 (60)
Social support from coaches	9 (45)
Unrealistic expectations from coaches	6 (30)
Social support from family	11 (55)
Used sport as an outlet	12 (60)
Thoughts of gratitude and resiliency	12 (60)
Managing expectations	6 (30)

**Table 3 ijerph-19-06885-t003:** Mean ACSI-28 Subscale and Overall Scores.

Category	Mean Score ± SD (Range)
ACSI-28, Overall	49.5 ± 11.8 (24–66)
Coping with Adversity	6.6 ± 2.1 (2–10)
Peaking under Pressure	6.8 ± 2.8 (1–11)
Goal Setting/Mental Preparation	7.2 ± 3.0 (2–11)
Concentration	6.8 ± 2.7 (1–11)
Freedom from Worry	4.9 ± 2.9 (0–10)
Confidence and Achievement Motivation	7.7 ± 1.7 (4–10)
Coachability	9.5 ± 2.0 (5–12)

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
