# Peer review of "The Collegiate Athlete Perspective on Return to Sport Amidst the COVID-19 Pandemic: A Qualitative Assessment of Confidence, Stress, and Coping Strategies"

_ijerph, 2022, doi:10.3390/ijerph19116885_

Round 1

Reviewer 1 Report

The paper was well-written overall and provided some great insight into a relevant and important topic. Student-Athlete Mental Health has been a topic on the rise, but with the COVID 19 pandemic, it is has been even more concerning. The paper is timely and provides more depth to the previously conducted studies regarding SA mental health during the pandemic. 

Introduction - authors provided a well-written rationale for the study and provided ample sources to support their purpose. 

Method - The methodology seems appropriate. I think the participants section could be expanded to discuss how many student-athletes were contacted (i.e. how many qualified) and then how many completed the interviews. I think this might be important because student-athletes suffering from various mental health issues may be reluctant to participate in an interview. It is important to show the percentage of athletes who were contacted, then who agreed to participate and then who completed the interviews to have a better understanding of how representative the data is of the population as a whole, particularly with the issue of mental health potentially precluding participation. Also, how did you decide about data saturation? Did that have an impact on the number of participants chosen or were there more willing to participate, but you cut them off due to saturation? Additionally, some sample questions from the interview guide may be appropriate to include. I think a few trustworthy procedures were mentioned, but were there others employed, such as reflexive journals, member checking, etc. to help avoid bias when interviewing and/or coding results? 

Results -This section was well-presented and provided quotes and descriptions of the themes. It could be helpful to provide a visual/diagram of the lower and higher order themes as a whole (as each of the 3 major themes seemed to have about 3 lower order themes within the respective sections). The quantitative data was presented and integrated with the qualitative data well. I like the use of the previous work to justify cutoff score criteria. 

Discussion - This section was nicely constructed and clearly outlined how the findings were connected with previous literature and how the findings can be used practically. I think the limitations discussed were appropriate. There may be some additional suggestions for future directions based on these limitations that could be added. 

Overall, a very well conceptualized, constructed, and written paper. 

Author Response

The number of student-athletes who were contacted, agreed to participate in the study, and who completed the study, were included (Please see the first sentence of "3.1. Characteristics of the Study Population").

Data saturation was determined by the point at which additional responses did not add new content. Once data saturation was reached, additional participants were not sought for the study (See "2.2. Recruitment and Data Collection").

Sample questions from the interview guide were included (See "2.2. Recruitment and Data Collection"). 

No other methods were used other than independent interviews and coding, etc. and the use of the adjunctive ACSI survey.

Common interviewee responses were listed by theme and presented in a table (see Table 2). The frequencies of the common responses were calculated as per the request of another reviewer.

Additional suggestions for future research given the limitations of the study were provided (see "Limitations").

Reviewer 2 Report

General comments

The Collegiate Athlete Perspective on Return to Sport amidst the COVID-19 Pandemic: A Qualitative Assessment of Confidence, Stress, and Coping Strategies. The authors have done a great job of providing an informative and meaningful addition to the current study field.

The number of participants in the article is quite limited. However, it can be accepted due to the special conditions (Covid-19). In addition, Abstract, Introduction, Materials and Methods, Results, Discussion (Limitations), Conclusions sections are well written.

More information about participants, information should be given about the total number and percentage of participants in the study (e.g. included the number of participants “contacted for the study, agreed to participate, completed the interviews” and dropped out). In addition, as stated by the authors, the number of participants in the article is quite limited. There may be additional recommendations for future research based on these limitations.

Author Response

The number/percentage of participants who were contacted for the study, agreed to participate, completed the study, and dropped out, were provided (see “3.1. Characteristics of the Study Population”). 

Future directions for research were provided given the study limitations (see “Limitations”).

Reviewer 3 Report

The present study entitled "The Collegiate Athlete Perspective on Return to Sport amidst the COVID-19 Pandemic: A Qualitative Assessment of Confidence, Stress, and Coping Strategies" has useful information about the psychic of the student-athletes and the coping strategies to reduce the impact of Covid-19 pandemic, in everyday life and in the expectations on the sport.

The main issue concern is the small size sample and the one-dimensional sex as the authors report.

Comment

In the introduction, it would be very useful to mention the possible or not gender differences in the parameters studied

Author Response

A discussion of the impact of gender differences on student-athlete mental health was added to the Introduction (see third paragraph of Introduction).

Reviewer 4 Report

Dear Authors,

Congratulation for your article, it is of interest and well understandable, for the results of your interview we suggest you to highlight the most important responses in each topic in percentage of the subjects that respond, like that we will know what were the most adapted coping strategy and by witch percentage of the subjects, …  And like that it will be possible to compare with other studies like the Estonian study. Also, the response of one subject should be of interest but have not the same weight for your conclusions that the same response by 50% of the subjects. Maybe a table with the responses by order of responses in the 3 topics (motivation, anxiety, coping) should be elucidative and useful for future studies compare with your study?

ACSI 28, the most scored was coachability it should be of interest to discuss that. The relation between results in this questionnaire and results in the interviews like higher score and gratitude, and other should be highlighted by a correlation to show if the relation is significative or not.

We wish you success in your publication

Author Response

A table with common responses listed by theme was added (see Table 2 under section "3.2. Themes").

A discussion of the highest sub-scale score (Coachability) and lowest sub-scale score (Freedom from Worry) was added to the section "4.4. Secondary Outcomes." Given the small sample size, it is not feasible to provide a statistical comparison of ASCI-28 scores between different qualitative responses. As noted, future research may be able to elucidate correlations between quantitative and qualitative results.